# Drug Delivery Systems for the Treatment of Knee Osteoarthritis: A Systematic Review of In Vivo Studies

**DOI:** 10.3390/ijms22179137

**Published:** 2021-08-24

**Authors:** Francesco Manlio Gambaro, Aldo Ummarino, Fernando Torres Andón, Flavio Ronzoni, Berardo Di Matteo, Elizaveta Kon

**Affiliations:** 1Department of Biomedical Sciences, Humanitas University Pieve Emanuele, 20090 Milan, Italy; a.ummarino@gmail.com (A.U.); fernando.torres.andon@usc.es (F.T.A.); flavio.ronzoni@hunimed.eu (F.R.); elizaveta.kon@humanitas.it (E.K.); 2IRCCS Humanitas Research Hospital, Rozzano, 20089 Milan, Italy; berardo.dimatteo@gmail.com; 3Center for Research in Molecular Medicine & Chronic Diseases (CIMUS), Universidade de Santiago de Compostela, 15705 Santiago de Compostela, Spain; 4Human Anatomy Unit, Department of Public Health, Experimental and Forensic Medicine, University of Pavia, 27100 Pavia, Italy; 5Department of Traumatology, Orthopaedics and Disaster Surgery, First Moscow State Medical University (Sechenov University), 119991 Moscow, Russia

**Keywords:** osteoarthritis, knee, drug delivery, nanoparticles, microparticles, hydrogels, tissue engineering

## Abstract

Many efforts have been made in the field of nanotechnology to improve the local and sustained release of drugs, which may be helpful to overcome the present limitations in the treatment of knee OA. Nano-/microparticles and/or hydrogels can be now engineered to improve the administration and intra-articular delivery of specific drugs, targeting molecular pathways and pathogenic mechanisms involved in OA progression and remission. In order to summarize the current state of this field, a systematic review of the literature was performed and 45 relevant studies were identified involving both animal models and humans. We found that polymeric nanoparticles loaded with anti-inflammatory drugs (i.e., dexamethasone or celecoxib) are the most frequently investigated drug delivery systems, followed by microparticles and hydrogels. In particular, the nanosystem most frequently used in preclinical research consists of PLGA-nanoparticles loaded with corticosteroids and non-steroidal anti-inflammatory drugs. Overall, improvement in histological features, reduction in joint inflammation, and improvement in clinical scores in patients were observed. The last advances in the field of nanotechnology could offer new opportunities to treat patients affected by knee OA, including those with previous meniscectomy. New smart drug delivery approaches, based on nanoparticles, microparticles, and hydrogels, may enhance the therapeutic potential of intra-articular agents by increasing the permanence of selected drugs inside the joint and better targeting specific receptors and tissues.

## 1. Introduction

Knee osteoarthritis (OA) affects around 250 million people worldwide, making it one of the leading causes of disability [1]. However, this estimated prevalence is likely going to grow as a consequence of the increased incidence of obesity and population ageing. Our understanding of its etiology has grown with time; the initial model of osteoarthritis as a simple cartilage degenerative disease has been overcome by a more complex multifactorial model encompassing an interplay among mechanical, metabolic, and inflammatory factors [2]. The complexity of the etiology is mirrored by the indications found in the treatment guidelines of the most important orthopedic societies as the Osteoarthritis Research Society International (OARSI) [3], the American College of Rheumatology (ACR) [4], and the American Academy of Orthopedic Surgeons (AAOS) [5]. They recommend addressing knee OA at multiple levels moving from nonpharmacological approaches such as weight control and adequate sport activity to oral or topical pharmacological treatments, intra-articular (IA) corticosteroid injections, and to novel promising cellular therapies such as stem cells [6]. Nevertheless, due to the multifactorial etiology and progressive nature of this condition, all the above mentioned approaches aim at controlling symptoms until the severity of the disease mandates surgical intervention with joint replacement. A recent study investigating the satisfaction of patients with the current OA treatment algorithm found that 78% of the patients had uncontrolled chronic knee pain [7]. Reasons behind the failure of current approaches may stem from both the lack of patients’ compliance and the frequent side effects related to the use of systemic drugs. Therefore, intra-articular injections seem to be more promising due to their preferential local effect. After intra-articular injections, the amount of substance inside the joint may rapidly decrease (<1% of the initial concentration) already 48 h after the injection, as observed in a rodent model by Allen et al., who employed elastin-like polypeptides [8]. Hence, new efforts have been made to develop new drug delivery approaches to overcome the present limitations in the treatment of knee OA and to allow a sustained release of the drug inside the joint.

To achieve this goal, several strategies have been investigated; in particular, nanoparticles, microparticles, or hydrogels have been used as drug delivery systems [9]. A schematic representation of these three main categories is presented in Figure 1. According to the International Organization for Standardization (ISO) [10], nanoparticles (NPs) are materials with an external dimension or an internal structure in the nanoscale (ranging from 1 nm to 1000 nm), therefore representing the smallest particles in nature after chemicals (atoms and small molecules) [11]. To date, many different materials have been exploited to manufacture NPs [12]; liposome-based NPs are so far those that have gained the most success in the medical world. Indeed, up until 2016, 18 NP drugs were approved by FDA, 12 of which were liposome-based, and 95% of them were indicated for cancer treatment [13]. In addition, in the last year, the use of lipid nanosystems for the in vivo delivery of mRNA in COVID-19 vaccines marked a breakthrough, which may be applied to other diseases in the near future [14,15]. However, if we consider those nanosystems as candidates for the treatment of OA, polymer-based NPs outstand liposomes in the number of publications. Moreover, in order to overcome the limitations of natural polymers (rapid degradation or limited mechanical properties), synthetic polymers have been developed such as polyethylene glycol (PEG) and poly-lactic-glycolic acid (PLGA) [16]. In addition to NPs, microparticles (MPs), with a size range between 1 and 1000 μm, have been investigated as drug delivery systems, especially those presenting a size similar to erythrocytes (7.5 μm) [17]. Drug delivery systems in the microscale display a more prolonged drug release versus other systems with different sizes [9]. Indeed, a study observed that drug bioactivity was recorded even after 2 months after the IA injection of MP in a mouse knee [18]. The beneficial properties of these MPs led to their investigation in a phase III clinical trial [19]. Finally, another type of nanostructure is represented by natural or synthetic hydrogels that are composed by a three-dimensional network of cross-linked polymer chains. These hydrogels can be either directly loaded with the desired drug or even with microspheres and/or nanospheres as drug carriers [20] (Figure 1). The aim of the present systematic review is to provide the readers with a comprehensive summary on the aforementioned “biotechnological strategies” explored so far to deliver drugs within the osteoarthritic knee.

## 2. Methods

The present systematic review was written according to the Preferred Reporting items for Systematic Reviews and Meta-analysis (PRISMA) guidelines [21], whose flowchart is depicted in Figure 2. A comprehensive search of PubMed, Google Scholar, Cochrane, and EMBASE was performed using various combinations of the following keywords: “Knee osteoarthritis,” “Drug delivery,” “Microparticles,” “Nanoparticles,” “Hydrogels.” The last research was performed on 14 June 2021 and two independent reviewers conducted the research independently (F.M.G. and A.U.). All 360 articles obtained by the literature search were screened according to the following inclusion criteria: (i) employment of a drug delivery system for the treatment of knee osteoarthritis; (ii) in vivo study; and (iii) knee intra-articular (IA) route of administration. Exclusion criteria were: (i) in vitro studies or review articles; (ii) other delivery methods than intra-articular application (e.g., transdermal application), (iii) not focused on osteoarthritis; (iv) not written in the English language.

## 3. Results

### 3.1. Drug Delivery Systems

We included 45 studies in total (Figure 2): 19 (42%) dealt with nanoparticles to improve the delivery of the pharmacological agent whereas 16 studies (36%) used microparticles and 10 studies (22%) employed hydrogels. The distribution of studies according to the year of publication is provided in Figure 3.

#### 3.1.1. Nanoparticles

A summary of the main features of all the studies employing NPs is reported in Table 1. Among the 19 included studies using NPs, 68% exploited polymers [22,23,24,25,26,27,28,29,30,31,32,33,34], 16% carbon-based NPs [35,36,37], 5% liposomes [38], 5% metal-based NPs [39], and 5% employed zeolitic imidazolate framework-8 (ZIF-8) [33]. In the polymeric NPs group, 56% of the studies used the Poly Lactic-co-Glycolic Acid (PLGA), making the latter the most employed material. Among these studies, four out of nine studies did not use a pure form of PLGA-based NPs but a combination of PLGA with Polyethylene glycol (PEG). In addition to PLGA, materials such as Poly *N*-isopropylacrylamide (pNIPAM), Polyester amide (PEA), Poloxamer, Hyaluronic Acid, and/or Chitosan have been employed.

PLGA is a synthetic polymer composed by a combination of two monomers: polylactic acid (PLA) and polyglycolic acid (PGA). PLA has a slow degradation rate while PGA presents a fast degradation rate. Thus, taking advantage of the different properties of these two molecules and by choosing the ratio of PLA and PGA, the degradation profile of PLGA can be designed for particular purposes, e.g., a slow and continuous release of drugs inside the knee [40]. Interestingly, polycaprolactone (PCL) is a very tough polyester, allowing a very slow degradation rate (up to 2–4 years) compared to the previous polymers. This feature obviously gains particular interest in the treatment of chronic conditions such as knee OA [41]. On the other hand, polyethylene glycol (PEG), with a high cross-linking capacity compared to other polymers, can be engineered with different sizes allowing the modification of NP-hydrophobicity, permeability, elasticity, and also degradation rate [41]. PEGylation, a process of linking PEG to other polymers or drugs, has been widely used as a “mask” strategy to avoid interaction with proteins or cells, reducing the immunogenicity and increasing the half-life after intravenous administration [42].

Among natural polymers, Hyaluronic Acid (HA) and Chitosan (CS) have been broadly used for biomedical research. HA is an anionic non-sulfated glycosaminoglycan highly present in our body (i.e., connective, epithelial, and neural tissues) and is a major component of the extracellular matrix, able to affect cell proliferation and migration, thus having important implications in tissue healing and modulation of inflammation and angiogenesis. High molecular weight HA has been FDA-approved to treat osteoarthritis of the knee via intra-articular injection, while HA with different sizes has been used to engineer drug delivery strategies, such as nanoparticles or hydrogels. Instead, Chitosan is a natural polysaccharide derived from chitin with the advantage of having free amine groups available for chemical modification, thus offering the chance to enhance specific functional properties. In addition, it has a similar structure to human glycosaminoglycans, therefore providing a possible microenvironment for chondrocyte proliferation and extracellular matrix synthesis [43].

Moving to carbon-based NPs, three studies employed these particular constructs [35,37,44]; in particular one study used fullerene-based NPs, while the other two studies employed graphene oxide or carbon nanotubules. Among the least represented groups, we found liposome-based NPs, employing a combination of lipids and HA [38] and metal-based NPs employing gold [39]. Finally, one study dealt with NPs composed by metal organic frameworks (MOFs) made of Zinc and imidazolate [33]. Indeed, MOFs present specific features that make them suitable for drug delivery since they have high porosity, high mechanical stability, and also a pH-responsive decomposition [45].

Summarizing the observed in vivo effects of NPs, six studies described improved histological structure after treatment [23,25,26,35,38,39] and three studies showed an increase in the OARSI score compared to control [22,26,33]. Concerning the level of inflammatory mediators, a decrease in IL-1β [26,33], TNF-α [25,26], MMP-3, and MMP-13 [33,36] expression levels was documented in most studies. Moreover, an increase in collagen type II levels was reported in three studies [23,33,36]. The above-mentioned results were observed at an average follow up time of 5 weeks (range 1–12 weeks).

Concerning the animal model on which the effects of the drug-loaded NPs were analyzed, rats were employed in the majority of the studies (81%), followed by rabbits (10%), bovines (6%), and sheep (3%).

#### 3.1.2. Microparticles

All 16 studies employing microparticles (MPs) to deliver pharmacological compounds used polymer-based MPs. In particular, 12 papers (80%) used PLGA [46,47,48,49,50,51,52,53,54,55,56,57], making it the most popular material to assemble MPs. Instead of employing a pure form of PLGA, two studies used PLGA combined with either PVA or PEA. The remaining three studies used poly-ε-caprolactone (PCL), Chitosan, or Heparin-based MPs.

Summarizing the in vivo effects of MPs, three studies reported an improvement in histological structure [53,58,59], while two studies on humans reported a significant reduction in average daily pain (ADP) [48,49]. Concerning the level of inflammatory mediators, only one study [60] observed a decrease in IL-1β, IL-6, IL-17, TNF-α, ADAMTS-5, and MMP3 expression levels.

The above-mentioned results were observed at an average follow up time of 6.8 weeks (range 2–12 weeks).

Concerning the species, four studies were performed on humans (27%) and the remaining on animals, in particular rats (60%), dogs (12%), and rabbits (12%). A summary of the features of all the studies employing MPs can be found in Table 2.

#### 3.1.3. Hydrogels

Among the selected articles, only 11 studies employed hydrogels as the drug delivery system for the treatment of OA; 60% of them were published in the year 2020. The most commonly adopted material to assemble the hydrogel was polylactic acid (PLA).

Indeed, PLA is prepared starting from L-lactide monomers that are subject to ring-opening polymerization to form L-lactic acid oligomers (LAo). This polymer is then mixed with a gelatin solution and is ready to be loaded with drugs, as described by three studies [61,62,63]. Instead, one study employed a HA-based gel [64] and another one a combination of HA and dextran [65].

One study employed a poloxamer-based hydrogel, a copolymer composed by polypropylene glycol (PPG) and PEG [66]. The remaining five studies used respectively α-chondroitin sulfate-ethyl glycol (EG), PEG, agarose, Chitosan–glycerin–borax, and poloxamer [66,67,68,69,70].

Summarizing the in vivo effects of hydrogels, two studies reported a more favorable histological structure [64,71], and five studies showed an improvement in the OARSI score compared to control [61,62,68,69,70]. Concerning the level of inflammatory mediators, six studies [61,62,63,66,70,71] observed a decrease in IL-1β, IL-6, IL-17, TNF-α, ADAMTS-5, and MMP3 expression levels. The abovementioned results were observed in an average follow up time of 8.4 weeks (range 2–24 weeks).

Concerning the animal model, five studies employed rats, four rabbits, and only one study a canine model. A summary of the features of all the studies employing hydrogels is reported in Table 3.

### 3.2. Types of Pharmacological Agents Delivered

Among the 45 included studies, the most frequently loaded drug category was represented by corticosteroids (16 studies); indeed, eight studies employed triamcinolone acetonide, five dexamethasone, and three betamethasone. The second most employed drug category was represented by NSAIDs (nine studies); in particular, three studies employed lornoxicam, two indomethacin, one ibuprofen, one naproxen, one diclofenac, and one piroxicam. Among the other molecules presenting a measurable pharmacological effect, hyaluronic acid, COX-2 specific inhibitors (coxib), chondroitin sulfate, clodronate, and KAFAK have been studied. Finally, the following drugs were employed only by a single study: methotrexate, adenosine, curcumin, oleic acid, NO-Hemoglobin Notch-1 siRNA, methyl prednisolone, antisense oligomers, eicosatetraenoic acid, rapamycin, and quercetin. Concerning the effects observed after treatment according to the drug used, the coxib group showed the most consistent improvement; all the studies employing either etoricoxib [29] or celecoxib [31,38] reported a better histological outcome after treatment. Instead, focusing on the IA concentration change in cytokines, a decrease in IL-1β, IL-6, TNF- α, MMP-3 was observed with all the main drug categories, i.e., with glucocorticoids [70], NSAIDs [60], and coxibs [29]. All the included studies on humans [48,49,51,54], employed triamcinolone acetonide. A graphic representation of the distribution of loaded drugs based on the type of nanomaterials is provided in Figure 4.

## 4. Discussion

Knowledge of the most commonly utilized drug delivery systems, such as nanoparticles, microparticles, or hydrogels, is crucial to understanding the direction of new therapeutic advances in the management of knee osteoarthritis. The present systematic review collected all the studies documenting the application of nano and microstructures as drug carriers in both animals and humans affected by knee OA. The main finding is that nanoparticles are the most employed drug delivery system, followed by microparticles and hydrogels. Moreover, the most frequently used materials to create these drug delivery systems are polymers and in particular PLGA. The most commonly loaded drug types are corticosteroids, followed by NSAIDs. Among the 45 studies included, only four [48,49,51,54] were performed in humans and all of these employed microparticles made of PLGA loaded with triamcinolone acetonide. These four trials were all published in the last three years, underlining that the application of novel intra-articular drug delivery system is facing the animal-to-human model transition, thus getting closer to a more consistent application in clinical practice. Multiple evidence shows why NPs are considered as the preferred drug delivery system: Firstly, the experience with NPs in other fields such as cancer therapy or vaccines has helped researchers to design NPs able to reach a particular tissue and specific cellular or molecular targets. In addition, the uptake of nanostructures by cells is higher compared to microstructures (1 and 10 µm) and NPs are commonly phagocytized more effectively than microspheres by macrophages [72]. Indeed, as macrophages are among the most important immune cells regulating the level of inflammation inside the knee joint, a proper uptake of the drugs by these cells (facilitated by NPs) can result in a reprogramming of their phenotype from an M1-like (which is pro-inflammatory) to an M2-like, which instead contributes to reduce the inflammatory distress in OA patients [73]. On the contrary, microspheres might be too big to be uptaken by macrophages and the preferential option for these microparticles is to release their content in the joint’s space, with a reduced ability to target specific receptors on the surface of precise immune cells. At the same time, hydrogels have a robust advantage versus other drug delivery systems, thanks to their possible application as biomaterials for bioprinting [74]. Moving toward the materials suitable for NP, MP, or hydrogel fabrication, the most employed were polymers, and in particular PLGA. The main features of this polymer are its synthetic nature and the possibility of modelling the drug releasing profile; the timing for degradation of PLGA-based NPs can be tuned to obtain a slower release by interfering with its molecular weight and ratio of lactide to glycolic acid in order to achieve the desired dosage and release interval depending upon the drug type [75]. The second most employed particles were liposomes. These carriers have the advantages of being nontoxic, flexible, biocompatible, completely biodegradable, and nonimmunogenic after systemic and non systemic administrations but they carry also some disadvantages such as a short half-life and high production cost [76]. Aside from the nature of the employed structures, many studies have investigated the ability of liposomes to produce a timely sustained drug release over time, and this is one of the main goals to achieve in the field of intra-articular injections for the treatment of knee OA. In regard to this aspect, hydrogels showed the longest observed beneficial effects in terms of reduction in cytokines concertation, better OARSI score, and histological features up to an average time of 8.4 weeks after treatment, compared to 6.8 weeks with MP and 5 weeks with NPs. Some important findings and preliminary trends can be garnered from this review, revealing that as a whole the literature in this area is still limited and a significant knowledge gap in the field exists.

## 5. Limitations

The main limitations of the present review are represented by the heterogeneity of the included studies in terms of materials (Table 1, Table 2 and Table 3), loaded drugs (Figure 4), analyzed outcomes, time intervals of observation, and animal models, thus revealing the need for further studies to identify the best performing drug delivery system.

## 6. Conclusions

The awareness that the “right” intra-articular carrier may optimize and increase the therapeutic effects of drugs has boosted the research on NPs, MPs, and hydrogels. The inherent translational potential in orthopaedics appears relevant considering both the clinical and socioeconomic burden of knee OA and the expanding range of therapeutic agents currently being tested on humans.

## Figures and Tables

**Figure 1 ijms-22-09137-f001:**
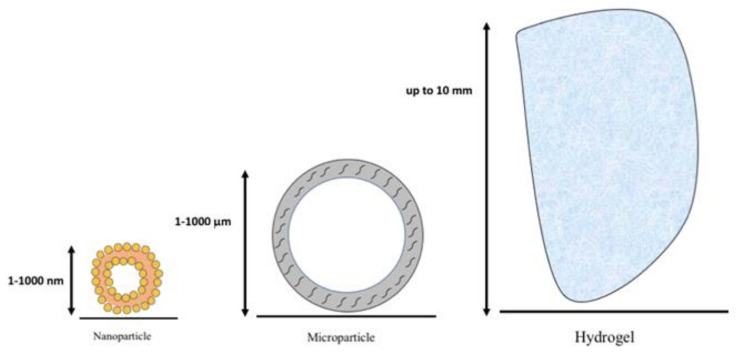
Schematic representation of the sizes of the three main types of drug delivery systems: the nanoparticle (size range 1–1000 nm), the microparticle (up to 10 μm), and the hydrogel (up to 10 mm). NOTE: the images are not in real proportions.

**Figure 2 ijms-22-09137-f002:**
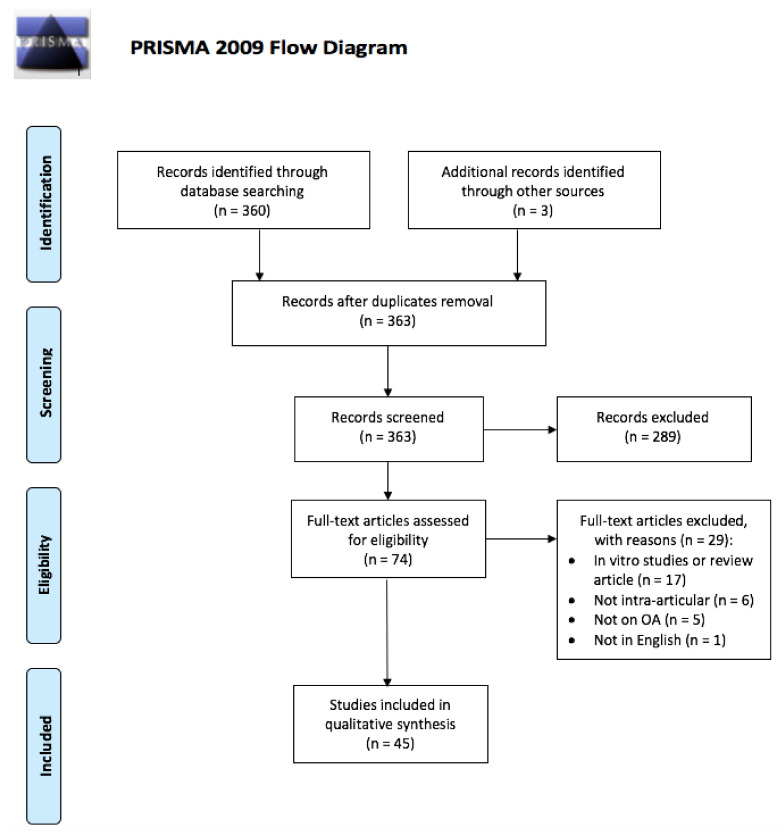
PRISMA Flowchart summarizing the papers’ selection process. Legend: OA: osteoarthritis.

**Figure 3 ijms-22-09137-f003:**
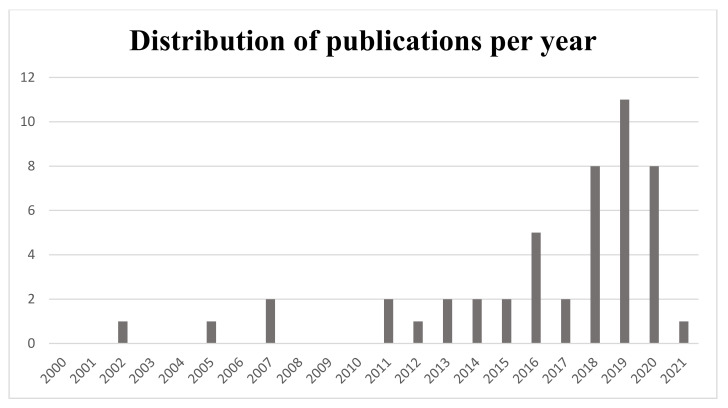
Distribution of the included studies according to the year of publication.

**Figure 4 ijms-22-09137-f004:**
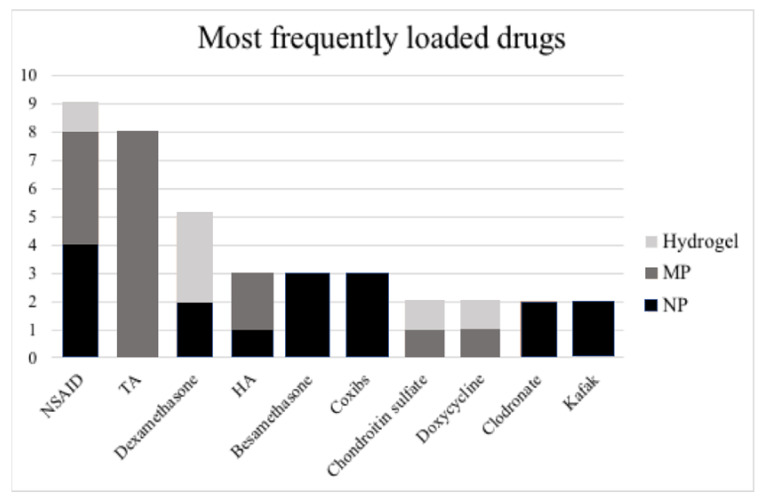
Distribution of drugs loaded on the different types of delivery systems in the included studies. Legend: HA—Hyaluronic acid, NSAID—non-steroidal anti-inflammatory drug; TA—Triamcinolone Acetonide.

**Table 1 ijms-22-09137-t001:** Synopsis of studies employing nanoparticles (NPs). Legend: ADAMTS-5—A disintegrin and metalloproteinase with thrombospondin motifs 5 2; CrmA—cytokine response modifier A; DPPC—1,2-dipalmitoyl-sn-glycero-3-phosphocholine; HA—hyaluronic acid; HDI—hexamethylene diisocyanate; IL—interleukin; MMP—matrix metalloproteinase; NGF—nerve growth factor; OARSI—Osteoarthritis Research Society International; PAE—poly(β-amino ester); PEAs—poly(ester amide)s; PEG—poly(ethylene glycol); PLA—poly(lactic acid): PLGA—poly(lactic-co-glycolic acid); pNIPAM—poly(*N*-isopropylacrylamide); TNF-α—tumor necrosis factor alpha; WBC—white blood cells; ZIF-8—zeolitic imidazolate framework-8.

Table	Articles	Drug	Material	Animal Model	Results	Ref
**Polymeric NPs**	Fan W 2018	Kartogenin	PEG-HDI-N-BOC Serinol	Rat	Better histological and OARSI score at 12 weeks compared to control	[22]
	Higaki M 2005	Betamethasone	PLGA	Rat	Decrease in inflammatory cells after 7 days	[23]
	Horisawa E 2002	Betamethasone	PLGA	Rabbit	Decreased joint swelling for 21 d	[24]
	Kamel R 2016	Indomethacin	Self assembling PLGA	Rat	↓ diameter; favorable histology; ↓ TNF-α in serum	[25]
	Kang C 2020	Curcumin	Acid-activable PAE	Rat	↓ TNF-α and IL-1β production, favorable histology	[25]
	Kang M 2016	Diclofenac/Kartogenin	Pluronic	Rat	↓ of OARSI score	[26]
	Kim SR 2016	Piroxicam	PLGA + Eudragit RL	Rat	Prolonged retention into joint compared to NPs without Eudragit RL	[27]
	Liu P 2019	Etoricoxib	PLGA/PEG	Rat	Favorable μCT; ↓ MMP-13 and ADAMTS-5; ↑ collagen and aggrecan	[28]
	Liu X 2019	Adenosine	PLGA-PEG	Rat	↓ OARSI score	[30]
	Villamagna IJ 2019	Celecoxib	PEAs	Sheep	↓ joint effusion; ↓ WBC	[31]
	Zerrillo L 2019	HA	PLGA	Rat	NP are still in the knee after 35 days	[32]
	Zhou F 2015	Berberine chloride	Chitosan	Rat	Higher anti-apoptosis activity and prolonged i.a. drug retention	[33]
	Zhou PH 2018	CrmA	HA and Chitosan	Rat	↓ OARSI score; ↓ IL-1β, MMP-3, MMP-13; collagen conserved	[34]
**Liposomes**	Dong J 2013	celecoxib	Liposome + hyaluronic acid	Rabbit	Favorable histology	[38]
**Carbon-based NPs**	Yudoh K 2007	KAFAK	Fullerene	Rabbit	Favorable histology	[35]
	Liu A 2019	Hyaluronan conjugation	Graphene oxide	Rat	↓ MMP-3 concentration in the joint	[36]
	Sacchetti C. 2014	Antisense oligomers	Carbon nanotubes	Rat	Inhibition of protein synthesis in chondrocytes and reduction in inflammation	[37]
**Metal-based NPs**	Sarkar A 2019	Fish oil protein, both in DPPC liposomes	Gold	Rat	↓ IL-1β, IL-12, PGE2, TNF-α	[39]
**Other NPs**	Zhou F 2020	S-methylisothiourea Catalase Anti-CD16/32	ZIF-8 (MOF)	Rat	Favorable histology and X-ray	[33]

**Table 2 ijms-22-09137-t002:** Synopsis of studies employing microparticles (MPs). Legend: ADAMTS-5—A disintegrin and metalloproteinase with thrombospondin motifs 5 2; CrmA—cytokine response modifier A; DPPC—1,2-dipalmitoyl-sn-glycero-3-phosphocholine; FBR—foreign-body responses; HA—hyaluronic acid; HDI—hexamethylene diisocyanate; IL—interleukin; MMP—matrix metalloproteinase; NGF—nerve growth factor; OARSI—Osteoarthritis Research Society International; PAE—poly(β-amino ester); PEAs—poly(ester amide)s; PEG—poly(ethylene glycol); PLA—poly(lactic acid): PLGA—poly(lactic-co-glycolic acid); TNF-α—tumor necrosis factor alpha; WBC—white blood cells.

Articles	Drug	Material	Animal Model	Results	Ref
Abou Elnour M 2019	Triamcinolone acetonide (TA)	PLGA	Rat	MP performed higher in inflammation suppression, compared to the free drug suspension	[46]
Allah HA 2016	Lornoxicam	Chitosan	Rat	Persistent inhibition of knee swelling and lower IL-6 levels until 14 days	[35]
Aydin O 2015	Doxycycline (D) and doxycycline-chondroitin sulfate (D-CS)	poly-ɛ-caprolactone (PCL)	Rabbit	Radiographic scores of D MS and D-CS MS groups improved after 8 weeks when compared to OA groups	[58]
Bodick N 2018	Triamcinolone acetonide	PLGA	Dog	Toxicity study, the synovial FBR to PLGA microspheres was focal and transient	[47]
Chen H 2019	Hydroxychloroquine	PLGA	Rat	Drug detectable inside the joint up to week 2	[50]
Conaghan GP 2018	Triamcinolone acetonide	PLGA	Human	Drug detectable inside the joint up to week 12	[48]
Conaghan GP 2018	Triamcinolone acetonide	PLGA	Human	Drug detectable inside the joint up to week 13	[49]
Ho Jin M 2019	Triamcinolone Acetonide Microcrystals	PLGA	Rat	The novel MS was physicochemically stable, with no changes in drug crystallinity and release profile over 12 months	[52]
Kraus VB 2017	Triamcinolone acetonide	PLGA	Human	Drug detectable inside the joint up to week 12	[51]
Kumar A 2015	Triamcinolone acetonide	PLGA FX006	Rat	Improved histological joint scores and better gait (pain) scores	[53]
Park JW 2016	Ibuprofen	PLGA, PVA	Rat	Reduction in IL-1β, IL-6, IL-17, TNF-α, ADAMTS-5 and MMP3	[60]
Rudnik-Jansen I 2019	triamcinolone acetonide	PLGA-PEA	Rat	A single intra-articular injection of TAA-loaded PEA microspheres reduced joint swelling and induced longer pain relief compared to bolus injection.	[55]
Russel SJ 2018	Triamcinolone acetonide	PLGA	Human	In diabetic patients, who represented the study population, intra-articular PLGA-TA provided better post-injective glycemic control than standard TA	[54]
Tellier EL 2018	TNF-α stimulated gene-6 (TSG-6)	Heparin based MP	Rat	After 21 days, cartilage thickness, volume, and attenuation were significantly increased	[59]
Zhang Z 2011	Lornoxicam	PLGA	Rat	Longer retention in the joint with MP system	[56]
Zhang Z 2012	Lornoxicam	PLGA	Rat	Retention of drug in the joint for >96 h	[57]

**Table 3 ijms-22-09137-t003:** Synopsis of studies employing hydrogels. Legend: ADAMTS-5A—disintegrin and metalloproteinase with thrombospondin motifs 5 2; ADH—adipic acid dihydrazide; CrmA—cytokine response modifier A; DPPC—1,2-dipalmitoyl-sn-glycero-3-phosphocholine; EG—ethyl-glycol; FBR—foreign-body responses; HA—hyaluronic acid; HDI—hexamethylene diisocyanate; IL—interleukin; MMP—matrix metalloproteinase; NGF—nerve growth factor; OARSI—Osteoarthritis Research Society International; PAE—poly(β-amino ester); PEAs—poly(ester amide)s; PEG—poly(ethylene glycol); PLA—poly(lactic acid): PLGA—poly(lactic-co-glycolic acid); TNF-α—tumor necrosis factor alpha; WBC—white blood cells.

Articles	Drug	Material	Animal Model	Results	Ref
Garcia-Fernandez L 2020	Naproxen or Dexamethasone	Dextran and HA	Rabbit	Dexamethasone group showed higher collagen type II levels and better recovery	[65]
Hui JH 2007	Chondroitin-sulfate	α-Chondroitin Sulfate-EG	Rabbit	Thicker layer composed of hyaline and fibrocartilage compared to control (saline)	[67]
Lu HT 2013	Doxycycline	HA hydrogel	Rabbit	Lower grade of OA in the study group compared to control (saline)	[64]
Matsuzaki T 2014	Rapamycin	Polylactic acid, gelatin	Rat	Delayed OA progression was maintained even at 16 weeks	[61]
Mok SW 2020	Quercetin	PEG hydrogel	Rat	Released of Que could be sustained for >28 days. Higher OARSI score than control	[68]
Stefani RM 2020	Dexamethasone	Agarose hydrogel loaded with MP	Dogs	Improved OARSI scores for proteoglycan, chondrocyte, and collagen pathology	[69]
Tanaka T 2019	Simvastatitn	Polylactic acid gelatin	Rat	Decreased level cartilage-degrading enzymes and IL-1β and increased level type II collagen	[62]
Tsubosaka T 2020	Eicosapentanoic acid	Polylactic acid gelatin	Rat	MMP-3-, MMP-13-, IL-1β-, and *p*-IKK α/β-positive cell ratio were significantly lower	[63]
Wang QS 2020	Dexamethasone	Chitosan–glycerin-borax–hydrogel	Rat	Decreased OA scores and joint inflammation compared to control	[70]
Zhang T 2020	Glucosamine	Poloxamer 407 and 188 hydrogel	Rabbit	Decreased swelling and inflammatory factors compared to control	[66]

## Data Availability

All the data retrieved have been included in the manuscript.

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
