# Peer review of "Drug Delivery Systems for the Treatment of Knee Osteoarthritis: A Systematic Review of In Vivo Studies"

_ijms, 2021, doi:10.3390/ijms22179137_

Round 1
Reviewer 1 Report
Please see the attachment.

Author Response
In this review authors focused on interesting topic of drug delivery systems and they potent usage in OA treatment. I find this manuscript interesting, but some corrections must be made before publication. Firstly, please correct English: some sentences are blurred other are colloquial.
- Thank you for your comment. We had the manuscript revised by a native English speaker in order to correct typos, remove redundancies and improve readability.
Technical issues:
Please change graphics (figures) to gray scale (yellow looks really unprofessional ).
- Figures changed as requested.
Figure 3 is placed in wrong section: it visualizes facts described in introduction – so place it in introduction section, or remove this figure, because that kind of visualization is not needed (it’s good described in the main text).
- We replaced the Figure 3 (now renumbered Fig. 1) immediately after the Introduction. We also changed the structure of the same figure to meet the suggestion of another reviewer who noted that y axis could be misleading for readers.
If authors put more than 2 references in the brackets they should be placed in order from smallest number to highest number not randomly – please correct the brackets.
- References have been ordered in the brackets, as requested, throughout all the manuscript.
Meritor issues:
In methods section: authors write about inclusion criteria; what about exclusion criteria? On the graphic I see how many papers were excluded from further analysis – it would be good to know why they were excluded.
- We added the exclusion criteria in the Materials and Methods and we also improved Fig. 2 (Prisma Flowchart) by including the reasons why some articles were excluded after full text reading.
In text fragment describing Table 1 and animal models used authors write rats rabbits but sheep – one sheep? Or sheeps. On contrary in Table 1 itself: model should be “rat” but sometimes authors put “rats” – correct in the table; in the same table in description of experiment by Sarkar –“ reduction of inflammation” measured with what method? Levels of proinflammatory cytokines external symptoms like swelling or pain, reddening?
- Corrections done as requested in the Tables. Please also note that the plural of “sheep” is “sheep” (same spelling for singular and plural! The word “sheeps” is not commonly used). With regard to “reduction of inflammation”, as correctly pointed out by the reviewer, pro-inflammatory cytokines were measured and therefore we better specified this aspect in the table.
Similar note I have to Table 2:” good effect until” what kind of result is this? Please state correctly the effect of experiment; similar with ”doable in DM” this is not on result of treatment, please state correctly effect of experiment;
- We better clarified the results of those studies: we meant to highlight the permanence of the drug inside the joint, so we rephrased accordingly.
Section 3.2 in cytokines should be beta in IL-1 ?
- Correct! We amended the text accordingly.
Reviewer 2 Report
Introduction
"In any case, after intra-articular injections the drug is not anymore detectable in the knee joint (<1% of the initial concentration) already 48h after the procedure [8]." I would suggest expanding this citation- it should be noted that it is based on the rat model and maybe it would be valuable to note what drugs were tested in this paper.
Methods
Figure 3. Arrows on y axis suggest linear values. This reviewer suggest removal of the arrows and/or adding a note that diameter axis increments are not linear.
Discussion
I would advise structurizing this section, exclude limitations of the study and final conclusion to separate (sub-)sections. Shorter, less complex phrases would improve understanding of authors' message to readers.
Short list of abbreviations used by authors would facilitate reading to readers less involved in recent publications.
Author Response
"In any case, after intra-articular injections the drug is not anymore detectable in the knee joint (<1% of the initial concentration) already 48h after the procedure [8]." I would suggest expanding this citation- it should be noted that it is based on the rat model and maybe it would be valuable to note what drugs were tested in this paper.
- Details added as per the reviewer’s request.
Methods
Figure 3. Arrows on y axis suggest linear values. This reviewer suggest removal of the arrows and/or adding a note that diameter axis increments are not linear.
- We agree that y axis may be misleading for reader so we decided to change the Figure so that it could be better interpreted. Obviously, we could not maintain the real proportions between nanoparticles, microparticles and hydrogels and we specified that in the figure legend.
Discussion
I would advise structurizing this section, exclude limitations of the study and final conclusion to separate (sub-)sections. Shorter, less complex phrases would improve understanding of authors' message to readers.
- We revised the text to correct typos and to improve the quality of the English language, with the aim of making it more fluent for the reader. As suggested by the reviewer, we create the sub-sections “Limitations” and “Conclusion”.
Short list of abbreviations used by authors would facilitate reading to readers less involved in recent publications.
- Thank you for the suggestion. A list of abbreviations was added to the text, just ahead of the Introduction.